# The Clinical Utility of the Saliva Proteome in Rare Diseases: A Pilot Study for Biomarker Discovery in Primary Sclerosing Cholangitis

**DOI:** 10.3390/jcm13020544

**Published:** 2024-01-18

**Authors:** Elisa Ceccherini, Elena Michelucci, Giovanni Signore, Barbara Coco, Michela Zari, Massimo Bellini, Maurizia Rossana Brunetto, Antonella Cecchettini, Silvia Rocchiccioli

**Affiliations:** 1Institute of Clinical Physiology, National Research Council, 56124 Pisa, Italy; elena.michelucci@pi.iccom.cnr.it (E.M.); giovanni.signore@unipi.it (G.S.); antonella.cecchettini@unipi.it (A.C.); silvia.rocchiccioli@ifc.cnr.it (S.R.); 2Institute of Chemistry of Organometallic Compounds, National Research Council, 56124 Pisa, Italy; 3Biochemistry Unit, Department of Biology, University of Pisa, 56123 Pisa, Italy; 4Hepatology Unit, Reference Centre of the Tuscany Region for Chronic Liver Disease and Cancer, University Hospital of Pisa, 56124 Pisa, Italy; b.coco@ao-pisa.toscana.it (B.C.); maurizia.brunetto@unipi.it (M.R.B.); 5Gastrointestinal Unit, Department of Translational Research and New Technologies in Medicine and Surgery, University of Pisa, 56124 Pisa, Italy; michela.zari@uslsudest.toscana.it (M.Z.); massimo.bellini@unipi.it (M.B.); 6Department of Clinical and Experimental Medicine, University of Pisa, 56126 Pisa, Italy

**Keywords:** primary sclerosing cholangitis, saliva proteomics, LC-MS/MS, biomarkers, disulfide-isomerase A3, peroxiredoxin-5

## Abstract

Background: Primary sclerosing cholangitis (PSC) is a rare chronic inflammatory liver disease characterized by biliary strictures and cholestasis. Due to the lack of effective serological indicators for diagnosis and prognosis, in the present study, we examined the potentiality of the saliva proteome to comprehensively screen for novel biomarkers. Methods: Saliva samples of PSC patients and healthy controls were processed and subsequently analyzed using a liquid chromatography–tandem mass spectrometry technique. A bioinformatic approach was applied to detect the differentially expressed proteins, their related biological functions and pathways, and the correlation with the clinical evidence in order to identify a possible marker for the PSC group. Results: We identified 25 differentially expressed proteins in PSC patients when compared to the healthy control group. Among them, eight proteins exhibited area under the curve values up to 0.800, suggesting these saliva proteins as good discriminators between the two groups. Multiple positive correlations were also identified between the dysregulated salivary proteins and increased serum alkaline phosphatase levels and the presence of ulcerative colitis. Pathway analysis revealed significant enrichments in the immune system, neutrophil degranulation, and in the interleukine-17 signaling pathway. Conclusion: We demonstrated the potentiality of saliva as a useful biofluid to obtain a fingerprint of the pathology, suggesting disulfide-isomerase A3 and peroxiredoxin-5 as the better discriminating proteins in PSC patients. Hence, analysis of saliva proteins could become, in future, a useful tool in the screening of patients with suspected PSC.

## 1. Introduction

Primary sclerosing cholangitis (PSC) is a rare chronic inflammatory disease of the biliary epithelium leading to bile duct strictures that are responsible for cholestasis and cirrhosis [1]. PSC is closely associated with ulcerative colitis, which is present in approximately 70% of PSC patients, exhibiting a 10-fold greater risk of developing colorectal cancer [2,3]. Analyzing data from the Italian National Rare Diseases Registry, 502 PSC patients were identified within the Italian population of approximately 60 million [4]. PSC patients are generally asymptomatic at the time of diagnosis, which is usually performed following persistent abnormal liver function tests, such as increased alkaline phosphatase (ALP) and gamma-glutamyltransferase (GGT) levels [5,6]. However, these levels are normal in some patients [1,7]. Several pathological conditions could lead to secondary sclerosing cholangitis, whose features can replicate PSC, and the possibility of distinguishing either diseases can be challenging [7,8]. To date, the lack of screening tests and serological markers for a certain diagnosis requires the discovery of novel effective PSC biomarkers. In recent years, mass spectrometry-based proteomic analysis of biological fluids has attracted the attention of researchers for the identification of potential biomarkers in several liver diseases [9,10]; however, proteomic studies concerning PSC are scarce.

The potentiality of bile and urine mass spectrometry-based proteomics has been recently explored in distinguishing PSC from cholangiocarcinoma (CCA) and benign biliary disorders [11,12,13]. Although the deregulation of several proteins was useful in differentiating CC from PSC and other benign biliary disorders, no biomarker has been proposed for PSC characterization. Biliary proteomics has also been exploited to highlight potential prognostic biomarkers of PSC, identifying calprotectin and IL-8 as indicators of disease severity and prognosis [14]. Since IL-8 plays a key role in the development of primary biliary cirrhosis [15,16,17,18] and the progression of liver cirrhosis [19], its relevance as a prognostic biomarker could be ascribed to a broader group of hepatopathies and not exclusively to PSC. Recently, integrated proteomic analysis of bile and serum, conducted on patients with PSC and non-PSC, identified cystatin-C and retina and anterior neural fold homeobox protein 2 as potential blood-based markers for PSC diagnosis [20]. Given the paucity of proteomic studies based on biological fluids from PSC patients, the aim of this study was to analyze and compare the saliva proteomic profiles of PSC patients and healthy controls to discover proteins that could be used as novel markers for a non-invasive screening and diagnosis of PSC.

## 2. Materials and Methods

### 2.1. Reagents, Solvents and Materials

All the reagents, solvents, and materials used for sample processing and analysis in liquid chromatography–tandem mass spectrometry (LC-MS/MS) were previously reported [21].

### 2.2. Study Design and Patient Characteristics

Ten patients with confirmed PSC free of dysplasia or CCA were subsequently recruited at the University Hospital of Pisa (Hepatology Unit and Gastroenterology Unit), according to the established inclusion criteria between November 2019 and May 2021. A control group (*n* = 10) of healthy subjects was also subsequently recruited. PSC patients and healthy controls (6 male and 4 females for each group) were in a range within 18–70 years (median age of 49 years) and 27–65 years (median age of 41 years), respectively. A saliva sample was obtained from each subject, and at the time of sampling, no patient or control had a diagnosis of oral pathologies. At the time of saliva collection, patients showed a median value for alanine aminotransferase (ALT), aspartate aminotransferase (AST), and total bilirubin of 26.50 U/L (11–103), 29.50 U/L (13–99), and 0.47 mg/dL (0.23–1.98), respectively. The median GGT, ALP, albumin, and gamma globulins values were 55 U/L (21–231), 132 U/L (48–181), 4.18 g/L (3.9–4.7), and 20% (16.5–24.1), respectively. In total, 70% of patients (7/10) had concomitant ulcerative colitis, 30% of patients (3/10) were positive to perinuclear anti-neutrophil cytoplasmic antibodies (P-ANCA), and 60% of patients (6/10) were positive to anti-nuclear antibodies (ANA). Among them, three patients were positive to ANA and P-ANCA. All the patients recruited had a well-compensated liver disease (Child Pugh A); none had signs of portal hypertension.

The study obtained the approval of the Comitato Etico Area Vasta Nord Ovest (Protocol code 57532, 30 October 2019, Pisa, Italy). All the patients signed an informed consent to participate in the study.

### 2.3. Salivary Sample Collection and Processing

Spontaneous saliva samples of PSC patients and healthy controls were collected and enriched in extracellular vesicles (EVs) by differential centrifugation isolation and protein extraction [22,23]. For each sample, 100 µg of proteins was subsequently reduced (using dithiothreitol 10 mM for 30 min, at 65 °C), alkylated (using iodoacetamide 20 mM for 30 min, at room temperature in dark conditions), and digested using trypsin (*w*/*w* ratio 1:50) at 37 °C for 16 h. Tryptic digestion was stopped with 1% trifluoroacetic acid for 45 min at 37 °C. The peptide mixtures were centrifuged and desalted according to our previous paper [23]. Each sample was resuspended in CH3CN/0.1% HCOOH (ratio 5/95) to achieve a final peptide concentration of 2 µg/µL and analyzed by LC-MS/MS.

### 2.4. LC-MS/MS Analysis

Samples were analyzed according to our previous paper [21] using a micro-HPLC (Eksigent Ekspert microLC 200, AB Sciex, Concord, ON, Canada) coupled with a Triple TOF 5600 mass spectrometer (AB Sciex, Concord, ON, Canada). As described before, protein quantification was achieved through SWATH-MS (Sequential Window Acquisition of all THeoretical fragment ion Mass Spectra) methodology. Only a few variations were adopted: the sample pool, analyzed with an information dependent acquisition (IDA) method, was injected in triplicate; the IDA elution program was extended (0 min, 5% B; 2 min, 5% B; 134 min, 50% B; 134.5 min, 90% B; 138.5 min, 90% B; 139 min, 5% B; 145 min, 5% B); the IDA MS/MS experiments were characterized by a dynamic exclusion duration of 15 s after 2 occurrences; the data independent acquisition SWATH-MS (DIA-SWATH-MS) elution program was slightly extended (0 min, 5% B; 2 min, 5% B; 62 min, 50% B; 62.5 min, 90% B; 66.5 min, 90% B; 67 min, 5% B; 73 min, 5% B), and the SWATH MS1 scan accumulation time was 250 milliseconds (overall duty cycle of 2.8 s, ~15 points per elution peak). The fifty Q1 variable windows used in the DIA-SWATH-MS method are shown in Appendix A.

### 2.5. Data Processing and Statistical Analysis

SWATH raw files were uploaded in the free universal software DIA-NN (version 1.8), and data processing was performed as already described [21]. A total of 733 proteins was quantified, and for each one, the fold change (FC) was calculated as the ratio between the mean of abundance in PSC patients and those in the control group. Proteins were considered differentially expressed (DEPs) when FC ≤ 1/1.5 or FC ≥ 1.5. GraphPad Prism 9 software was used to perform a Shapiro–Wilk test assessing the normal distribution of the data, and a two-tailed Mann–Whitney U test corrected with the Bonferroni method for multiple comparisons was applied to consider the significant differences in protein levels between the patients and controls. SPSS 29.0.1.0 statistical software was used to perform a point-biserial correlation to verify the possible correlation between the continuous variables and the dichotomous ones and for the receiver operating characteristic (ROC) analysis to evaluate the sensitivity and specificity of each protein marker in PSC. For all analyses, a *p* value < 0.05 was considered statistically significant. The DEPs were also subjected to Gene Ontology (GO) enrichment analysis using the David annotation tool, and pathway analysis using the Reactome Database. Enrichment analyses were performed using Fisher’s exact test, and terms with a *p* value < 0.01 were considered significant [24].

## 3. Results

### 3.1. Differentially Expressed Proteins (DEPs) in PSC Patients

The impact of PSC in salivary protein expression was assessed by comparing the protein levels in PSC patients and a control group. A total of 25 proteins were found to be significantly dysregulated in the PSC group (Figure 1); among them, 22 were upregulated (Figure 1a–d), and the serum amyloid P-component, tumor-associated calcium signal transducer 2, and dynein axonemal assembly factor 1 were downregulated (Figure 1e). We focused on the DEPs to investigate in detail their role in the pathologic mechanisms of PSC and their potential as diagnostic biomarkers.

### 3.2. Functional Network Analysis for DEPs

To evaluate the functional networks of the significant DEPs, a GO enrichment analysis was carried out using David-Functional Annotation Tools with respect to the biological processes (BPs), the cellular components (CCs), and the molecular functions (MFs). As reported in Table 1, nine BP GO terms were statistically enriched, natural killer cell degranulation (*p* = 3.9 × 10^−5^) and chaperone-mediated protein complex assembly (*p* = 4.3 × 10^−4^) being the most significant.

As expected, in CCs the extracellular exosomes emerged, including 20/25 of the DEPs (*p* = 5.5 × 10^−14^). Six significantly enriched terms were highlighted in MFs, and the most enriched was MHC class II protein complex binding (*p* = 7.8 × 10^−4^). To more deeply elucidate the involvement of DEPs in PSC mechanisms, a pathway analysis was performed using the Reactome Database. As reported in Table 2, the most significantly enriched pathways were related to the immune system (innate immune system, *p* = 1.8 × 10^−7^; neutrophil degranulation, *p* = 1.7 × 10^−7^) and inflammatory responses (IL-17 signaling pathway, *p* = 7.3 × 10^−4^). Moreover, we also highlighted the significant enrichment in metabolic processes regarding carbohydrates (*p* = 2.6 × 10^−3^) and vitamins and cofactors (*p* = 6.2 × 10^−3^).

### 3.3. Correlation between DEPs and Clinical Features of PSC Patients

To evaluate the possibility that DEPs could represent an indicator of biliary damage and/or the presence of ulcerative colitis, we applied a score = +1 in presence of ulcerative colitis and/or when the level of ALP was out of range (ALP value ≥ 105 g/dL), for the opposite, we applied a score = 0; then, a point-biserial analysis was performed. Applying an R ≥ 0.500 and R ≤ −0.500 as cut-off values, we highlighted several significant positive correlations between DEPs and the clinical features of PSC patients mentioned above. In particular, nicotinamide phosphoribosyltransferase (R = 0.515, *p* = 0.020) and hyaluronan synthase 1 (R = 0.531, *p* = 0.016) exerted significant correlations with the presence of ulcerative colitis. Additional correlations were shown between ALT levels and the salivary amount of core histone macro-H2A.1 (R = 0.678, *p* = 0.001) and immunoglobulin lambda variable 3-19 (R = 0.583, *p* = 0.007). The closest correlation was found between the core histone macro-H2A.1 and the increase in ALP levels.

### 3.4. ROC Curves Analysis

To evaluate the specificity (true negative rate) and sensitivity (true positive rate) of each DEP as a potential salivary marker in PSC, we conducted a ROC analysis in which the area under the curve (AUC) represents the potentiality of each protein in distinguishing between the PSC group and the control group. The accuracy of the AUC was categorized as follows: excellent (0.9–1), good (0.8–0.899), fair (0.7–0.799), poor (0.6–0.699), and not useful when the AUC value was <0.6. The results of the performance analysis highlighted that 8 out of 25 DEPs had an AUC ≥ 8 (Figure 2). The turquoise lines represent the sensitivity and specificity of each protein in distinguishing the PSC group from the control group, and the green diagonal line represents a purely random performance. Among the DEPs, disulfide-isomerase A3 showed the highest AUC value (AUC: 0.900, 95% Confidence Interval (CI) 0.763–1.000), followed by peroxiredoxin-5 (AUC: 0.865, 95% CI 0.679–1.000) and nicotinamide phosphoribosyltransferase (AUC: 0.850, 95% CI 0.663–1.000).

## 4. Discussion

PSC has several clinical manifestations; in particular, it is characterized by inflammation, fibrosis, and strictures of bile ducts. The lack of screening tests and serological markers for a certain diagnosis require the discovery of novel effective PSC biomarkers. To date, available proteomic studies have been conducted on bile, serum, and urine samples from PSC patients [11,13,14,20,25,26,27], but none of them have investigated the potentiality of saliva as a useful biofluid in biomarker research. In the present study, we explored LC-MS/MS proteomics to identify potential salivary markers for PSC. To this end the salivary EV-enriched proteomic profiles of PSC patients and healthy subjects were compared, and 25 proteins exhibited significantly different levels between the two groups (Figure 1). To facilitate the selection of candidate protein markers, a ROC analysis was performed, and 8 out of 25 proteins attained an AUC value ≥ 0.8 (Figure 2). Among them, the disulfide-isomerase A3 (AUC: 0.900, 95% CI 0.763–1.000) and peroxiredoxin-5 (AUC: 0.865, 95% CI 0.679–1.000) exhibited the best capability to discriminate the PSC group. Protein disulfide-isomerase A3 is an endoplasmic reticulum enzyme, which catalyzes the formation and isomerization of disulfide bonds in newly synthesized glycoproteins [28]. Published data concerning its involvement in PSC are scarce. For example, Gonzalez-Sanchez and colleagues detected high levels of disulfide-isomerase A3 in the small and large bile duct samples from a patient with PSC as well as in the ATP-binding cassette transporter B4 knockout mice, used as a model of chronic cholangiopathy [29]. Interestingly, a recent proteomics study showed that disulfide-isomerase A3 was present in the bile of patients with PSC and not detected in the choledocholithiasis patients used as the control group [26]. It is interesting to note that a proteomic study conducted on the intestinal mucosa derived from a mice model of ulcerative colitis highlighted the downregulation of disulfide-isomerase A3 compared to the levels detected in normal tissue [30]. Indeed, approximately 70% of PSC patients have concomitant ulcerative colitis [3]; thus, disulfide-isomerase A3 could play a key role as a specific biomarker for PSC diagnosis. Peroxiredoxin-5 is an enzyme whose expression is regulated by inflammatory stimuli such as hydrogen peroxide, peroxynitrite, and alkylhydroperoxide [31]. Several studies have demonstrated increased expression of peroxiredoxin-5 in mouse bone marrow-derived macrophages upon stimulation with pro-inflammatory stimuli such as interferon-γ and lipopolysaccharides [32,33]. Interestingly, recent studies found that in PSC patients, the peribiliary epithelium was already infiltrated by macrophages in the early stage of the disease [34,35]. Macrophage recruitment to the biliary environment has been also confirmed in murine models of PSC (Mdr2−/− mouse), where they promote injury and cholestasis [34]. These results are also corroborated by data derived from our pathway analyses. In Table 2, we reported the immune system (*p* = 1.8 × 10^−7^) and the IL-17 signaling pathway (*p* = 7.3 × 10^−4^) as the most significant deregulated pathways in PSC patients compared to the control group. Indeed, IL-17 is a pro-inflammatory cytokine that participates in several fibrotic diseases, including liver fibrosis [36], inducing macrophage infiltration and local inflammatory response [37,38]. According to this evidence, our data might indicate that increased levels of salivary disulfide-isomerase A3 and peroxiredoxin-5 could be potential peripheral indicators of macrophage-mediated biliary fibrosis in PSC patients. ROC analysis revealed additional proteins with potentially good discriminatory power for pinpointing possible patients suffering from PSC disease (Figure 2), and among them, nicotinamide phosphoribosyltransferase (R = 0.515, *p* = 0.020) and hyaluronan synthase 1 (R = 0.531, *p* = 0.016) are also positively correlated with the presence of ulcerative colitis. Although the correlation between the increased expression of these two salivary proteins and the presence of ulcerative colitis does not imply causation, additional sporadic studies have shown significant upregulation of nicotinamide phosphoribosyltransferase [39] and hyaluronan synthase 1 [40] in intestinal biopsies from patients with inflammatory bowel disease, corroborating our correlation analysis. Last but not least, we highlighted positive correlations between the increased ALP serum levels (ALP value ≥ 105 g/dL), which represents an indicator of biliary injury, and the salivary amount of core histone macro-H2A.1 (R = 0.686, *p* < 0.001) and immunoglobulin lambda variable 3–19 (R = 0.511, *p* = 0.021). Currently, there are no studies that have investigated the expression of these proteins in tissue biopsies or biological fluids of PSC patients. Nevertheless, an increased liver content of core histone macro-H2A.1 has been identified in murine models of hepatic steatosis [41] and fat-associated hepatocellular carcinoma [42], and an upregulation of immunoglobulin lambda variable 3-19 has been shown in the serum samples of 16 patients with autoimmune cirrhosis compared to the healthy control group [43]. Although saliva is not considered functionally associated with the bile ducts, some of the identified DEPs, such as disulfide-isomerase A3, exhibited a clear deregulation in the bile and serum of PSC patients [20], confirming saliva as potential biological fluid in PSC characterization. To the best of our knowledge, this is the first salivary proteomics study based on PSC patients and healthy subjects in which we succeeded in demonstrating the potentiality of saliva as a useful biofluid in biomarker discovery, suggesting disulfide-isomerase A3 and peroxiredoxin-5 as discriminating proteins between the PSC group and healthy subjects. Several other proteins positively correlated with increased ALP levels and ulcerative colitis presence were also identified in this study, but determination of their clinical significance requires additional investigational efforts. This is a pilot study with intrinsic limitations related to the restricted cohort of patients that could negatively impact our statistical analysis and, consequently, the clinical significance of the identified potential biomarker for PSC diagnosis and research. PSC is a rare disorder; thus, multi-center studies involving a larger number of patients are needed to validate our results. Even if ultracentrifugation is currently used as the gold standard method for the isolation of EVs from biological fluids [44], this technique could be implemented with more effective methods. Although our results are preliminary, the use of saliva as diagnostic biofluid and the identified proteins could become a helpful strategy for suspected PSC confirmation in the future.

## Figures and Tables

**Figure 1 jcm-13-00544-f001:**
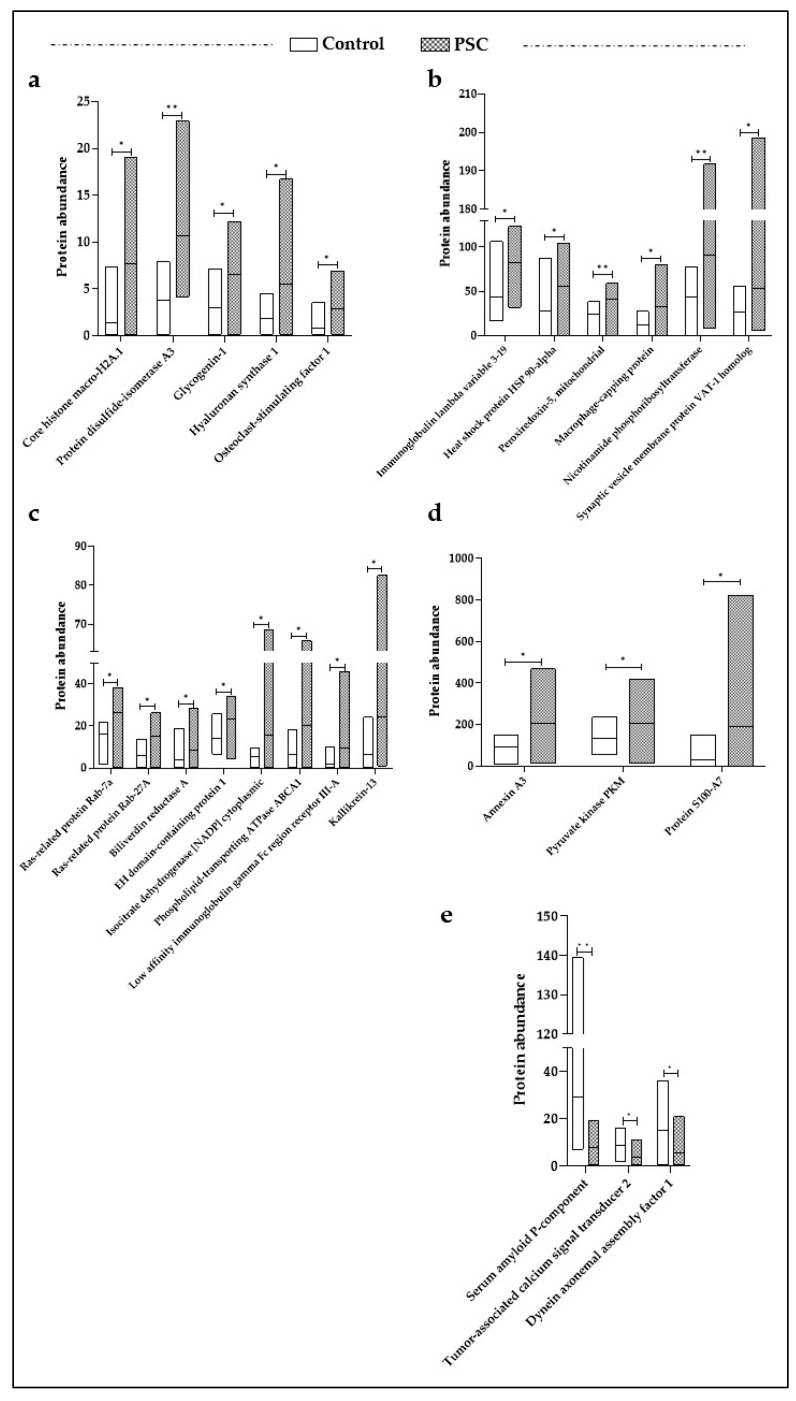
Protein abundances for significant DEPs in PSC patients. The up-regulated proteins are shown in panel (**a**–**d**), and the down-regulated ones are shown in panel (**e**). A two-tailed Mann–Whitney U test and Bonferroni multiple comparison test were performed to consider the significant differences in protein expression between the PSC patients and healthy controls (* *p* value < 0.05, ** *p* value < 0.01).

**Figure 2 jcm-13-00544-f002:**
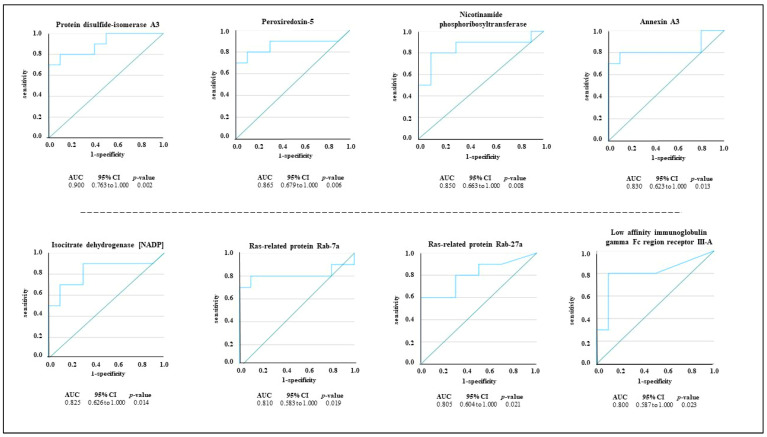
Receiver operating characteristic (ROC) curves for DEPs in PSC patients. Proteins exhibiting an AUC ≥ 8 are reported, and for each one, the 95% CI and *p* value are reported. The turquoise lines represent the sensitivity and specificity of each protein in distinguishing the PSC group from the control group, and the green diagonal line represents a purely random performance.

**Table 1 jcm-13-00544-t001:** Functional annotation analysis performed with David tools. Enrichment analyses were performed using Fisher’s exact test, and the Gene Ontology (GO) terms exhibiting a *p* value < 0.01 were considered significant. For each GO category (Biological Processes, Cellular Components, and Molecular Functions), the enrichment terms, the number of proteins, and the *p* value are reported.

	Number of Proteins	*p* Value
Biological Processes		
natural killer cell degranulation	2	3.9 × 10^−5^
chaperone-mediated protein complex assembly	2	4.3 × 10^−4^
positive regulation of interferon-beta production	2	1.5 × 10^−3^
protein folding	3	2.1 × 10^−3^
protein secretion	2	3.1 × 10^−3^
endosomal transport	2	4.9 × 10^−3^
positive regulation of protein catabolic process	2	7.1 × 10^−3^
cholesterol homeostasis	2	9.0 × 10^−3^
Cellular Components		
extracellular exosome	20	5.5 × 10^−14^
Molecular Functions		
MHC class II protein complex binding	2	7.8 × 10^−4^
cadherin binding	4	1.1 × 10^−3^
NADP binding	2	1.8 × 10^−3^
GTP binding	4	2.8 × 10^−3^
oxidoreductase activity	3	4.3 × 10^−3^
calcium ion binding	5	4.5 × 10^−3^

**Table 2 jcm-13-00544-t002:** Pathway analysis performed using Reactome Databases. Enrichment analyses were performed using Fisher’s exact test, and terms exhibiting a *p* value < 0.01 were considered significant. For each term, the number of proteins and the *p* value are reported.

	Number of Proteins	*p* Value
Innate immune system	12	1.8 × 10^−7^
Neutrophil degranulation	9	1.7 × 10^−7^
IL-17 signaling pathway	3	7.3 × 10^−4^
Metabolism of carbohydrates	4	2.6 × 10^−3^
Metabolism of vitamins and cofactors	3	6.2 × 10^−3^
Cellular responses to stress	5	1.8 × 10^−2^

## Data Availability

Data will be made available on request.

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
