# Peer review of "The Clinical Utility of the Saliva Proteome in Rare Diseases: A Pilot Study for Biomarker Discovery in Primary Sclerosing Cholangitis"

_jcm, 2024, doi:10.3390/jcm13020544_

Round 1

Reviewer 1 Report

Comments and Suggestions for Authors

The present study examined the potentiality of the saliva proteome to comprehensively screen for novel biomarkers. The results showed that disulfide-isomerase A3 and peroxiredoxin-5 as the better discriminating proteins in PSC patients. Hence, the measurement of saliva proteins could become in future a useful screening of patients with suspected PSC. The study was overall well conducted and of application potentials. Some minor points are listed as below.

1. The sample size was relatively small and the results should be validated in larger cohorts.

2. Multi-center trials would help verify the efficacy and safety of the method.

Comments on the Quality of English Language

Minor editing of English language required

Author Response

Dear reviewer,

we agree with your observations and aware that this preliminary study has the limitation of the cohort analyzed and that it is necessary to validate our results in a larger population recruited in a multicenter study. Indeed, these limitations were stated at the end of the discussion to circumstantiate our results, hoping that they could be the starting point to support research by other clinical and research centers.

Reviewer 2 Report

Comments and Suggestions for Authors

The research by Ceccherini et al. examined the potentiality of the saliva proteome in PBC patients in finding potential diagnostic biomarkers. The conclusions drawn are a good starting point for further analyses (especially molecular analysis like RNA-seq or similar). Some sections, like the introduction and discussion, are hard to read and need improvement. Please rearrange this section for better understanding and make it easier to follow the researcher's thoughts. The discussion must be written more consistently. It should also be indicated which information in the literature is consistent and which is contrary to the obtained results. Also, the discussion section needs to include the part that discusses the limitations of this study. 

Because PSC mainly affects young men, I have concerns about the amount and age of patients selected for this study. The range of 18 to 74 years is too much age dispersion. PSC is a rare disease, but authors should perform analysis in higher groups of patients to discuss potential diagnostic or prognostic biomarker implications. 

Moreover, there is no information about the disease stage of patients arranged for this study. Were the patients during some treatment, or was the saliva collection also correlated with the first or subsequent visit or diagnosis?

 Figure 1 needs to be improved. In their present form, they are hard to understand and read. The authors have a possibility of y-axis shortening Diurng working with GraphPad. The authors have an option for y-axis shortening to better present their results. 

The authors need to be precise when using the "biomarker" definition. It needs to be clarified which kind of biomarker authors have in mind - diagnostic, prognostic, or therapeutic. Please explain this in the discussion. 

Has a dentist confirmed the condition of the patient's oral cavity? Did they have any oral or periodontal disease?

Methods: Did authors immediately analyze saliva samples after collection or store them at -80C? 

Line 63 - moreover, IL-8 could also be used to diagnose different cholestatic disorders - PBC [PMID: 36982376]. Please add this information to the introduction section. 

Moreover, I recommend changing the form of publication from article to communication. In its current form, the article requires additional analysis and thorough discussion.

Comments on the Quality of English Language

Moderate editing of English language required. 

Author Response

Dear Reviewer,
thank you for valuable suggestions that greatly improved the quality of the manuscript.

  • Some sections, like the introduction and discussion, are hard to read and need improvement. Please rearrange this section for better understanding and make it easier to follow the researcher's thoughts. The discussion must be written more consistently. It should also be indicated which information in the literature is consistent and which is contrary to the obtained results. Also, the discussion section needs to include the part that discusses the limitations of this study. 

Thank you for these suggestions. We have revised the entire manuscript to make it clearer.

  • Because PSC mainly affects young men, I have concerns about the amount and age of patients selected for this study. The range of 18 to 74 years is too much age dispersion.

Patient were recruited consecutively at University Hospital of Pisa. Median age was 49 years, and only 2 patients were older than 50 years. These data were added to the manuscript.

  • PSC is a rare disease, but authors should perform analysis in higher groups of patients to discuss potential diagnostic or prognostic biomarker implications. 

We agree with your observations and aware that this preliminary study has the limitation of the cohort analyzed, thus it is necessary to validate our results in a larger population recruited in a multicenter study. These limitations were stated at the end of the discussion to circumstantiate our results, hoping that they could be the starting point to support further research in order to elucidate their potential diagnostic or prognostic biomarker implications.

  • Moreover, there is no information about the disease stage of patients arranged for this study. Were the patients during some treatment, or was the saliva collection also correlated with the first or subsequent visit or diagnosis?

All the patient recruited had a well-compensated liver disease (Child Pugh A); none had signs of portal hypertension. No curative established treatment for PSC is available at the moment. Ursodeoxycholic acid (UDCA) is licensed for PSC in many Europe countries, including Italy. All the patients were under UDCA therapy (dose 15-20 mg/kg). Saliva collection was performed during a follow up visit after the PSC diagnosis, and all patients were 8 hours fasting.

  • Figure 1 needs to be improved. In their present form, they are hard to understand and read. The authors have a possibility of y-axis shortening Diurng working with GraphPad. The authors have an option for y-axis shortening to better present their results. 

We have revised Figure 1 to make it more comprehensible.

  • The authors need to be precise when using the "biomarker" definition. It needs to be clarified which kind of biomarker authors have in mind - diagnostic, prognostic, or therapeutic. Please explain this in the discussion. 

At the end of the introduction, we specified the aim of the study, and explained our purpose in more detail along the discussion.

  • Has a dentist confirmed the condition of the patient's oral cavity? Did they have any oral or periodontal disease?

No patient had clinical signs of active periodontal disease at the saliva collection.

  • Methods: Did authors immediately analyze saliva samples after collection or store them at -80C? 

Saliva samples were processed immediately since the presence of bacteria would degrade them.

  • Line 63 - moreover, IL-8 could also be used to diagnose different cholestatic disorders - PBC [PMID: 36982376]. Please add this information to the introduction section. 

As you suggested, we revised the introduction adding important information on the role of IL-8 in PBC and liver cirrhosis.

  • Moreover, I recommend changing the form of publication from article to communication. In its current form, the article requires additional analysis and thorough discussion.

For this request we prefer to leave the decision to the Editor.

Reviewer 3 Report

Comments and Suggestions for Authors

This manuscript was aimed to evaluate the clinical utility of saliva proteome in  primary sclerosing cholangitis. This study is very interesting and gives a new information for future study.

I have a few questions.

1. Is there any data related with the prognosis of patients (response of steriod tx.) or type of PSC?

2. How about serum protein? Is there any data? If not, you can add some comments in discussion section.

3. You can present using volcano plot or other plots instead of the Fig 1.

Comments on the Quality of English Language

English was well presented.

Author Response

Dear Reviewer,
thank you for valuable suggestions that greatly improved the quality of the manuscript.

  1. Is there any data related with the prognosis of patients (response of steriod tx.) or type of PSC?

The study protocol approved by the ethics committee did not include prognostic information and, therefore, we are unable to provide this data.

  1. How about serum protein? Is there any data? If not, you can add some comments in discussion section.

As suggested, we added information on albumin (median value of 4.18 g/L) and gamma globulins (median value 20%) levels in the manuscript.

  1. You can present using volcano plot or other plots instead of the Fig 1.

We have revised Figure 1 to make it more comprehensible.

Reviewer 4 Report

Comments and Suggestions for Authors

The study on the potentiality of saliva proteome as a source of biomarkers for Primary Sclerosing Cholangitis (PSC) is well-structured and informative. However, here are some critical comments:

1: Sample Size and Selection: The study includes a small sample size (10 PSC patients and 10 healthy controls). While PSC is a rare disease, a larger sample size would strengthen the statistical power and reliability of the findings.

2: Control Group Characteristics: The characteristics of the control group are not fully described. It would be beneficial to include information on age, gender, and other relevant factors to ensure a proper comparison with the PSC group.

3: Saliva Proteome Specificity: The study highlights the potentiality of saliva as a biofluid for PSC diagnosis, but it would be useful to discuss the specificity of saliva proteome markers in distinguishing PSC from other liver diseases or inflammatory conditions.

4: Statistical Analysis: The statistical methods are generally well-described, but additional details on the correction for multiple testing in the identification of differentially expressed proteins (DEPs) would enhance the robustness of the results.

5: Clinical Correlations: While the study identifies correlations between dysregulated salivary proteins and clinical parameters, the clinical relevance and the strength of these correlations should be discussed in more detail. Additionally, correlation does not imply causation, and this should be emphasized.

6: ROC Analysis Interpretation: The ROC analysis is informative, but the clinical significance of the AUC values should be discussed more comprehensively. Consideration of sensitivity, specificity, and positive predictive values would provide a more nuanced understanding of the diagnostic potential of the identified biomarkers.

7: Protein Identification and Validation: The study identifies 25 proteins with significantly different levels, but the identification and validation of these proteins should be discussed in the context of their functional relevance to PSC. Further studies, including independent validation cohorts, are essential to confirm the findings.

8: Study Reproducibility: Details on how the experiment can be reproduced by other researchers are crucial for the scientific community. Providing information on protocols, reagents, and data availability would enhance the reproducibility of the study.

9: Limitations: The study should discuss its limitations, such as the cross-sectional design, the limited sample size, and potential confounding factors, to provide a balanced interpretation of the results.

10: Future Directions: Discussing potential future directions and applications of the findings, such as the development of diagnostic assays or the integration of saliva proteomics with other diagnostic modalities, would add depth to the conclusion.

Comments on the Quality of English Language

Moderate editing of English language required

Author Response

Dear Reviewer,
thank you for valuable suggestions that greatly improved the quality of the manuscript.

1: Sample Size and Selection: The study includes a small sample size (10 PSC patients and 10 healthy controls). While PSC is a rare disease, a larger sample size would strengthen the statistical power and reliability of the findings.

We agree with your comments and are aware that, in this preliminary study, the limitation of the cohort analyzed might affect the statistical significance. These limitations have been indicated at the end of the discussion to substantiate our results, hoping that they can be the starting point to support the research in validating the results in a larger cohort.

2: Control Group Characteristics: The characteristics of the control group are not fully described. It would be beneficial to include information on age, gender, and other relevant factors to ensure a proper comparison with the PSC group.

As suggested, we added the information of the control group for proper comparison with the PSC group.

3: Saliva Proteome Specificity: The study highlights the potentiality of saliva as a biofluid for PSC diagnosis, but it would be useful to discuss the specificity of saliva proteome markers in distinguishing PSC from other liver diseases or inflammatory conditions.

This suggestion was very valuable in improving the discussion of the manuscript. We added new information regarding the specificity of identified proteins in distinguishing PSC from other conditions affecting liver or inflammatory bowel disease.

4: Statistical Analysis: The statistical methods are generally well-described, but additional details on the correction for multiple testing in the identification of differentially expressed proteins (DEPs) would enhance the robustness of the results.

Thank you for identifying this oversight, we have added the information in the section “2.5 Data processing and statistical analysis”.

5: Clinical Correlations: While the study identifies correlations between dysregulated salivary proteins and clinical parameters, the clinical relevance and the strength of these correlations should be discussed in more detail. Additionally, correlation does not imply causation, and this should be emphasized.

This suggestion was very valuable in improving the discussion of the manuscript. We discussed the correlations obtained in more details, highlighting the clinical relevance of the identified proteins and the related bibliography.

6: ROC Analysis Interpretation: The ROC analysis is informative, but the clinical significance of the AUC values should be discussed more comprehensively. Consideration of sensitivity, specificity, and positive predictive values would provide a more nuanced understanding of the diagnostic potential of the identified biomarkers.

According to these suggestions, we revised the section “3.5 ROC curves analysis” adding more information to better understand the diagnostic potential of the identified biomarkers.

7: Protein Identification and Validation: The study identifies 25 proteins with significantly different levels, but the identification and validation of these proteins should be discussed in the context of their functional relevance to PSC. Further studies, including independent validation cohorts, are essential to confirm the findings.

We revised the manuscript discussing the relevance of proteins in PSC and specifying the need to validate our results in a larger cohort, enrolled in multi-center studies due to the rarity of PSC.

8: Study Reproducibility: Details on how the experiment can be reproduced by other researchers are crucial for the scientific community. Providing information on protocols, reagents, and data availability would enhance the reproducibility of the study.

To ensure the reproducibility of the study, in the "materials and methods" section, we described in detail all the reagents and protocols used for the isolation of EVs, protein extraction, preparation and analysis of peptide mixture by LC-MS/MS.

9: Limitations: The study should discuss its limitations, such as the cross-sectional design, the limited sample size, and potential confounding factors, to provide a balanced interpretation of the results.

Limitations of the study were included in the manuscript discussion to circumstantiate the results that need to be validated in a larger cohort. 

10: Future Directions: Discussing potential future directions and applications of the findings, such as the development of diagnostic assays or the integration of saliva proteomics with other diagnostic modalities, would add depth to the conclusion.

We added a final sentence about possible future implications of our results. However, given the nature of the study and its limitations, we feel it is speculative to add further comments.

Round 2

Reviewer 2 Report

Comments and Suggestions for Authors

Authors responded to all the comments satisfactorily. I recommend changing the form of publication from article to communication. 

Author Response

Authors responded to all the comments satisfactorily. I recommend changing the form of publication from article to communication.

Dear Reviewer,
thank you for your cooperation. As suggested, we changed the manuscript type to "communication". 

Reviewer 4 Report

Comments and Suggestions for Authors

The authors have addressed all the concerns.

Author Response

The authors have addressed all the concerns.

Dear reviewer,

thank you for your cooperation.